# Manufacturing of Food Packaging Based on Nanocellulose: Current Advances and Challenges

**DOI:** 10.3390/nano10091726

**Published:** 2020-08-31

**Authors:** Ghislain Fotie, Sara Limbo, Luciano Piergiovanni

**Affiliations:** DeFENS, Department of Food, Environmental and Nutritional Sciences, Università degli Studi di Milano, Via Celoria 2, 20133 Milano, Italy; sara.limbo@unimi.it (S.L.); luciano.piergiovanni@unimi.it (L.P.)

**Keywords:** hydrophobic nanocellulose, nanocellulose coating and lamination, bio-based, compostable food packaging

## Abstract

Nowadays, environmental pollution due to synthetic polymers represents one of the biggest worldwide challenges. As demonstrated in numerous scientific articles, plant-based nanocellulose (NC) is a biodegradable and nontoxic material whose mechanical, rheological, and gas barrier properties are competitive compared to those of oil-based plastics. However, the sensitivity of NC in humid ambient and lack of thermosealability have proven to be a major obstacle that hinders its breakthrough in various sectors including food packaging. In recent years, attempts have been made in order to provide a hydrophobic character to NC through chemical modifications. In addition, extensive works on nanocellulose applications in food packaging such as coating, layer-by-layer, casting, and electrospinning have been reported. Despite these enormous advances, it can easily be observed that packaging manufacturers have not yet shown a particular interest in terms of applicability and processability of the nanocellulose due to the lack of guidelines and guarantee on the success of their implementation. This review is useful for researchers and packaging manufacturers because it puts emphasis on recent works that have dealt with the nanocellulose applications and focuses on the best strategies to be adopted for swift and sustainable industrial manufacturing scale-up of high-performance bio-based/compostable packaging in replacement of the oil-based counterparts used today.

## 1. Introduction

### 1.1. Cellulose Properties and Capabilities

First revealed in the world by a French scientist Anselme Payen in 1838, cellulose is the most abundant biopolymer on the earth. Cellulose is generally synthesized by plants, but it is also available in some bacteria (*Acetobacter xylinum*) [1]. d-glucopyranose molecules contribute to the building blocks of cellulose polymer chains. Anhydroglucose units are linked by β(1–4) glycosidic bonds and two units of anhydroglucose form an anhydrocellobiose structure (Figure 1).

Cellulose content varies according to the botanic specie, e.g., the cotton has about 90% cellulose, wood 40–50%, or bast fibers such as flax, hemp, or ramie about 70–80% cellulose. Cellulose chains are tough, fibrous, and water-insoluble structures arranged in microfibrils with high strength and other superior mechanical properties. Intermolecular bonds provide stiffness, while the intramolecular bonds provide sheet structures to the polymer chain [2]. Succinctly, apart from paper and boards, cellulose without any modification does not have many applications in packaging and, in particular, in flexible packaging materials. However, when the cellulose is subjected to chemical, mechanical, or enzymatic modifications, it can be employed for example as coatings on food and packaging. Such modifications, which include the obtainment of methylcellulose (MC), hydroxypropyl cellulose (EC), and carboxymethyl cellulose (CMC) have long been used as coatings on foods, emulsifiers, bulking-extender agent, texturizers, and pharmaceutical excipient [3]. Furthermore, secondary cellulose acetate (56% acetyl groups) has been employed in thermoplastic processing and grafted cyclic lactones simultaneously onto polysaccharide and hydroxy functional plasticizer for packaging applications [4,5]. More recently, these cellulose derivatives have been employed in combination with other matrixes, i.e., lactoferrin and lysozyme have been successfully incorporated into a cellulose-based material for being used as an antimicrobial packaging for conservation of thin slices of raw meat [6]; furthermore hydroxypropylmethyl cellulose (HPMC) has been found to be very compatible with chitosan to set up a packaging displaying both antimicrobial and gas barrier properties [7,8]. In food packaging, a regenerated and biodegradable cellulose like the cellophane has long been used as a thin and transparent sheet for its high barrier to gases, oils, greases, and bacteria. Such properties have contributed to the cellophane expansion in complex multilayer packaging capable of preventing meat from oxidation and discoloration and from spoilage of fresh and dried oxygen-sensitive foods [9,10]. However, if seeking higher gas barrier biodegradable materials, cellophane that is even an expensive material is not appropriate to guarantee such requirements. The advent of the nanotechnology has opened a door for new opportunities to create cellulose-based materials with higher gas barrier performance. In recent years though, research on nanocellulose has sprung up and many scientific works dealing with studies of the chemicophysical properties and potential applications in various sectors including food packaging have been published. As a matter of illustration, when used as coating, layer-by-layer (LbL), casting or fillers, the nanocellulose displays unique gas/oil barrier and optical/mechanical properties useful for food packaging [11,12].

### 1.2. Introduction to Nanocellulose

Nanotechnology that implies the manipulation of very tiny matter has made possible the discovery of the nanocellulose (NC). According to ISO/TS 20477:201, NC is a structure whose at least one dimension is equal or less than 100 nm, and it can be subdivided into cellulose nanocrystals and cellulose nanofibrils, which are chemically and mechanically produced, respectively [13]. Throughout the years, the nomenclature of the cellulose nanocrystals has changed from being called whiskers, needles to rod-like while the cellulose nanofibrils have been referred to nanofibrillated cellulose (NFC), nanofibrillar cellulose (NFC), microfibrillated cellulose (MFC), microfibrillar cellulose (MFC), cellulose microfibril (CMF), and cellulose nanofiber (CNF). However, to avoid any confusion, in this review about plant-based nanocellulose, the acronym “CNCs” has been used to indicate the cellulose nanocrystals while both “MFC” and “CNFs” have been referred to cellulose nanofibrils [14,15]. However, it has to be pointed out that the ambiguity exists towards micro-/nanofibrils because the mechanical process breaks the cellulosic materials in both nano- and microparticles simultaneously.

Cellulose nanocrystals (CNCs) are very tiny and crystalline nanoparticles that are obtained by a chemical process through an acidic or oxidative hydrolysis during which the amorphous regions of the cellulosic materials are rapidly hydrolyzed releasing unblemished crystalline regions [16]. Unlike CNCs, the cellulose nanofibrils (MFC/CNFs) contain both amorphous and crystalline regions and they are prepared by mechanical shear during which the cellulose is defibrillated and disintegrated [17]. NC is characterized by its size, aspect ratio and shape, surface charges, tensile strength and stiffness, thixotropic behavior and crystallinity degree, which all depend not only on the type of production adopted but also on the raw material sources (plant-based) used. By comparison of their general features, CNCs are a more crystalline structure having a length of 100–250 nm and width of 3–50 nm while the CNFs have a higher aspect ratio with the length >1 µm and width 3–100 nm. Prior the “top-down” process (chemical or mechanical shear), it is necessary to subject recalcitrant raw materials to enzymatic, alkaline, and acid pretreatments to facilitate the disintegration of fibers in order to increase the production yield on the one hand and make the process less energy consuming on the other hand [18]. Widely reported in literature, nanocellulose has been demonstrated to exhibit an effective barrier to gases and aroma, thermal resistance, and mechanical and rheological properties useful for food packaging. However, as nanocellulose swells in a humid environment, this harms its exceptional properties and as a result, it hinders its breakthrough and application on an industrial scale. Accordingly, scientists have performed chemical functionalizations to provide hydrophobic character to nanocellulose and its subsequent resistance to water. In addition, in view of achieving better performance for food packaging applications, operational aspects and manipulation/handling of the NC requires very strong attention and expertise.

### 1.3. CNCs and MFC/CNFs Production

Acid hydrolysis is traditionally adopted to isolate the cellulose nanocrystals (CNCs) from the cellulosic sources. The propensity of the cellulose fibers to tightly adhere makes the crystalline regions less accessible to acid attack while amorphous regions are preferably accessible and hydrolyzed. Various acids such as sulfuric acid, hydrochloride acid, periodic acid, and phosphoric acid have been successfully employed for the extraction of CNCs from many cellulosic raw materials such as cotton linters, wood pulp, microcrystalline cellulose (MCC), soy husks, rice hulls, etc. The type of hydrolysis and raw material influence the CNCs properties like the morphology and size, the crystallinity, and the charges density. As a matter of fact, CNCs obtained by HCl do not contain charges on their surface while CNCs obtained by other acids contain residual charges attributable to their higher colloidal stability [19]. As a result, a wide spectrum of residual charges has been found of great benefits for the functionalization of the CNCs surface. Another widespread technique of extraction consists of using strong oxidizing agents such as the ammonium persulfate (APS) and the 2,2,6,6-tetramethylpiperidine-1-oxyl (TEMPO) for the CNCs production. Both APS and TEMPO can be used directly on the raw material or on CNCs thereby introducing carboxylic groups on the surface. By generating repulsive forces, such negative charges on the CNCs surface prevent the crystals from aggregating through electrostatic stabilization of the colloidal suspension [16,20].

Unlike CNCs, CNFs are generally manufactured by mechanical shear. More often preceded by enzymatic, chemical, or mechanical pretreatment, the mechanical disintegration of the cellulose to obtain CNFs, usually includes phases such as grinding, refining, and high-pressure homogenizing (Figure 2). Following refining that usually occurs when the dilute cellulosic fibers are forced to pass between two discs to be separated; the refined slurry of small fibers is subjected to high pressure through the homogenization [21]. After the homogenization, the concentration of the slurry is usually about 0.5–3% [22]. Due to this energy intensive process to produce the MFC/CNFs, cellulosic raw materials have to be subjected to mechanical, chemical, or enzymatic pretreatments to facilitate the next steps of the production. Such enzymatic (i.e., endoglucanase), acidic, and mechanical (i.e., sonication) pretreatments can also be used in combination [23]. In addition, a research related to MFC/CNFs production upon chemical pretreatments reported energy savings up to 98% [24].

It bears noting that the source of the raw materials, the type of pretreatment and mechanical disintegration dictate the production yield and the MFC/CNFs features such as crystallinity, morphology, and surface ratio [21]. Due to the presence of amorphous and crystalline regions in their structure, CNFs are known to be more flexible and less crystalline than the CNCs. APS and TEMPO have been also used to produce more stable colloidal suspensions of cellulose nanofibrils due to negative carboxylic charges present on their surface [25,26]. Both CNCs and MFC/CNFs exhibit gas, aroma, and solvent barrier properties required in food packaging for shelf-life extension. However, as pointed out above, humid conditions compromise the nanocellulose properties; therefore, the hydrophobization could allow for the incorporation of NC into packaging.

## 2. Chemical Modifications of Nanocellulose

Nanocellulose modification has been a subject of interest because it can allow to mitigate the water sensitivity that strongly hinders the implementation of cellulose-based nanocomposites in various fields of applications including food packaging. One major limitation to overcome lies in the poor compatibility and affinity of the nanocellulose with most hydrophobic and synthetic polymers like polyolefins [27]. Nanocellulose is by nature a very hydrophilic and hygroscopic material due to adjacent polar groups (-OH) linked by weak hydrogen bonds on the structure. Hence, the tendency of the nanocellulose to aggregate makes difficult its use in nonpolar solvents and blending with synthetic matrix like polymers. However, the abundance of reactive hydroxyl groups on the nanocellulose surface creates a wide spectrum of opportunities for chemical functionalizations [28,29]. Furthermore, adsorption of chemical compounds onto the cellulose nanocrystals surface can be either by “affinity” or by “electrostatic interaction” with surfactants or positive charges [30]. Succinctly, the final scope of the NC modification would be that of improving its dispersibility in water or solvents, providing high compatibility with hydrophic polymers and improving gases and water barrier properties [31].

### 2.1. Nanocellulose Oxidation

Generally, carboxylated nanocellulose occurs when oxidizing agents such as TEMPO, APS, and others are applied on cellulosic raw materials or uncharged nanocellulose (CNCs and CNFs) [16,32]. Oxidation is mostly performed on the already-produced nanocellulose with a catalytic amount of TEMPO using a secondary oxidant such as sodium hypochlorite or sodium chlorite during which the degree of substitution of the oxidation is increased by the use of sodium bromide [32]. Contrarily to TEMPO, APS preparation is usually performed on lignocellulosic raw materials during which simultaneous hydrolysis and oxidation take place. The main advantage of using APS is the fact that not only it can be used on recalcitrant raw materials with aromatic rings but it also contributes to bleaching the raw materials by providing white suspensions at the end of the reaction. Both APS and TEMPO make possible the introduction of the carboxylic groups on nanocellulose surface. In addition, aldehyde groups can also be found on the NC surface through the periodate oxidation by a selective cleavage of vicinal diols, in which the 2,3-diol of glucopyranose ring of the cellulose breaks down. The oxidized CNCs produced with that approach was used to improve the dry tensile index of 32.6% of the paper [33].

### 2.2. Esterification

The esterification of the cellulose nanocrystals has been extensively reported by many scientists. Ester compounds are formed when a carboxylic acid or acyl halides react with an alcoholic group. CNFs have been modified with oleic acid to graft esters groups on its surface to improve the compatibility with PLA (polylactic acid). After the blending of PLA matrix and esterified CNFs, the mechanical properties like tensile strength and Young’s modulus of nanocomposites were improved twice as compared to the pure PLA films. In addition, the barrier to water vapor was also improved on PLA blended with modified CNFs [34]. The first esterification of the CNCs has been implemented in situ in a single step with HCl in presence of mixed acids such as acetic acid and butyric acid, and it resulted that about half of hydroxyl was converted into esters at the end of the esterification [35]. De Castro et al. performed an esterification by surface grafting of CNCs with natural antimicrobial rosin using a green process called “sol-react” [36]. Similarly, acetylated CNCs have been prepared by using the acetic acid catalyzed by citric acid [37]. More recently, an in situ esterification was successfully performed by crystals interaction containing hydroxyl and carboxylate groups present on their surface. Several plastic films were coated with those esterified CNCs, resulting in significant improvement in oxygen barrier at 50% and 80% RH [38]. Esterified CNFs was also produced by the succinic anhydride esterification, yielding a better thermal stability and higher transparency compared to the neat one [39].

### 2.3. Amidation

The mechanism of amidation is formed by the involvement of the carboxylic acid reacting with a primary amine. The amidation of the carboxylated CNCs have been the first to be implemented by using a combination of 1-ethyl-3-(3-dimethylaminopropyl) carbodiimide (EDC) and *N*-hydroxysuccinimide (NHS) [40,41]. Cellulose nanocrystals have been modified by reacting citric acid-crosslinked CNCs with chitosan to produce amidated CNCs. Nanostarch films used with the amidated CNCs showed 230% tensile strength improvement and decrease of water vapor permeability and moisture absorption of 87.4% and 25.6%, respectively [42]. Amidation has been performed also on CNFs through the reaction between the *N*-hydroxysuccinimide-modified rhodamine B ester and nanofibers initially modified with amine groups using 4-(Boc-aminomethyl)phenyl isothiocyanate [43].

### 2.4. Carbamination

The carbamination has been implemented to provide hydrophobic behavior to the cellulose nanocrystals through a reaction between isocyanates and the hydroxyl groups of the cellulose nanocrystals. For example, the toluene-2,4-diisocyanate (TDI) has been used to attach polymers and molecules, and nonpolar isocyanates have been used to modify the nanocrystals properties [44]. Very recently, a work has been reported in which sisal cellulose nanocrystals were modified with n-octadecyl isocyanate without any catalysts and the hydrophobization of the cellulose nanocrystals by using isocyanates and grafting with phenyl isocyanates through TDI and trimethylamine in catalyzed conditions [45].

### 2.5. Grafting “from” and Grafting “onto”

“Grafting onto” approach typically involves attachment of presynthesized polymer chains onto nanocellulose surface by means of a coupling agent, whereas in “grafting from” approach, the polymer chains are formed in the course of the grafting process by in situ surface-initiated polymerization from immobilized initiators on the substrate. Both approaches have been used for polymers “grafting onto” CNCs or CNFs surface [16]. As such, the “grafting onto” technique allowed developing malted polypropylene-grafted cellulose nanofibrils. After the grafting, a significant reinforcement of the atactic polypropylene was observed while the layer of the grafted CNCs showed a significant decrease in mechanical properties [46]. “Grafting from” can be divided in the graft polymerization, the ring opening, and the radical polymerization. It has been reported a successful “grafting from” by grafting the polycaprolactone onto CNCs surface mediated by a ring-opening polymerization with a subsequent improvement in thermal stability and increase of hydrophobicity [47]. In another study, a cellulose-based hydrogel, poly(acrylic acid)-modified poly(glycidylmethacrylate)-grafted nanocellulose (PAPGNC) was synthesized by graft copolymerization reaction of glycidylmethacrylate onto nanocellulose in the presence of ethylene glycol dimethacrylate as cross-linker followed by immobilization of poly(acrylic acid). The results showed that PAPGNC was very efficient in adsorbing chicken egg white lysozyme from aqueous solutions [48].

## 3. Nanocellulose and Food Packaging

Scientists and researchers have been working on the process of incorporating the nanocellulose into the food packaging. As highlighted above, various ways of nanocellulose applications such as coating, casting, and layer-by-layer (LbL) have been developed.

### 3.1. Layer-by-Layer (LbL) Assembly

LbL assembly has been long employed to deposit a very thin layer of functional substances onto surfaces. Dispersions of cellulose nanocrystals are coated multiple times on solid substrates (several layers) until reaching a sufficient thickness able to provide gas barrier properties, mechanical properties, and wet-strength requirements [49,50]. Layers of multicomponent films can be coated on polymers or papers to form a complex mosaic of multistructures to keep unblemished properties that are useful for food packaging applications. It is worth mentioning that the electrostatic interaction can be exploited by alternating polyanionic and polycationic NC layers to obtain denser and ultrathin layers [51]. Polyaniline has been alternated with cellulose nanofibrils in an electrostatic layer-by-layer deposition process to form thin nanocomposite films [52]. In search of creating barrier and active packaging, several researchers have created LbL by interacting the nanocellulose with cationic polyelectrolytes such as the chitosan [53], poly(ethyleneimine), poly(allylamine hydrochloride) and poly(diallyldimethylammonium chloride) [54] or polyamidoamine wet-strength resin [55]. The combination of multiple layers shows high versatility for coating of end-use structures like containers, trays, and bottles [56].

### 3.2. Nanocomposite Extrusion

Extrusion is a generic process in which raw materials are melted at a given temperature and formed into a continuous profile to obtain a desired thickness. After, the molten matrix is then forced into a die, which shapes the polymer or composite into a shape that hardens during cooling. However, the incorporation of nanocellulose into plastic polymers is a challenge due to the lack of compatibility between both materials [57]. Recently, nanocellulose has been used as filler for the mechanical reinforcement of adhesives and polymers such as PLA taking advantage of the very high aspect ratio and specific surface area of the nanocellulose [58]. Esterified CNCs grafted with organic acid chlorides have been used in extrusion with low-density polyethylene. As a result, the mechanical properties like the elongation at break were significantly improved [59]. Nanocellulose/montmorillonite (MTM) composite films have been prepared from 2,2,6,6-tetramethylpiperidine-1-oxyl radical (TEMPO)-oxidized cellulose nanofibrils (TOCNs) with MTM nanoplatelets. The resulted composite films obtained were transparent and flexible exhibiting ultrahigh mechanical and oxygen barrier properties [60].

### 3.3. Electrospinning

Electrospinning (ES) is a technique based on the electrostatic forces of the fiber to produce fibers up to macrometric scale. The polymer or composite melt can be produced by heating from either resistance heating or circulating fluids. In the past years, cellulose-based materials have been successfully electrospun [61]. The alignment of the fibers that occurs during the ES, improves or changes the thermal/mechanical properties and crystallinity of the electrospun structure. Polyethylene and polyvinyl alcohol have been electrospun with the MFC and CNCs and the respective electrospun complex matrices displayed higher compatibility and improved mechanical properties [62,63,64,65]. Very recently, electrospun matrix processed by electrospinning coating between films of nanocellulose and polyhydroxyalkanoates was shown to exhibit water resistance. Martínez reported the nanocomposites fully synthesized by bacteria composed of polyhydroxybutyrate-co-hydroxyvalerate (PHBV) matrices reinforced with bacterial cellulose nanowhiskers (BCNW) by electrospinning technique and the nanocomposite obtained was highly dispersed exhibiting a reduction in oxygen permeability without relevant modifications in mechanical performance [66]. Very recently, poly(vinyl alcohol) (PVA) and cellulose nanocrystals (CNCs) composites have been prepared by the electrospinning method. After heat treatment at 170 °C for 2 h, composites showed a decrease in tensile strength, an increase in tensile stiffness, and a decrease in strain to yield (%). That effect was attributable to both cross-linking between -OH groups on PVA and esterification between vinyl groups and CNCs [67].

### 3.4. Casting and Evaporation

Casting is a generic operation of evaporation of water or solvent from the NC at a controlled temperature to obtain dried films of neat or modified NC. It is important to note that CNCs are brittle compared to flexible CNFs; therefore, the addition of the plasticizers like sorbitol and glycerol can mitigate the brittleness by reducing the capacity of the CNCs structure to form hydrogen bonds [55]. A nanocomposite material has been successfully obtained by the casting of a mixture formed of cellulose nanocrystals and plasticized starch. The casting structure underwent a remarkable enhancement of water resistance and mechanical properties such as tensile strength and Young’s modulus from 3.9 to 11.9 MPa and from 31.9 to 498.2 MPa, respectively [68]. Even though the casting is not commonly used in packaging, Wang and coworkers recently created polypropylene laminates based on nanocellulose (CNCs and CNFs) casting. Following the casting, CNCs films were found to be clearer and denser (≈1.4 g/m^2^ vs. ≈1.1) than CNFs films given the same thickness [69].

## 4. Nanocellulose and Resins Barrier: Properties, Applications, and Market Trends

By virtue of high barrier performance, thermoplastic resins barrier such as the copolymer of ethyl vinyl alcohol (EVOH), polyvinylidene chloride (PVDC), and aromatic polyamide (MXD6) have long been used in end-use industries for food shelf-life extension by blocking small molecules of gases such as CO_2_ and O_2_, water vapor and aroma. Compared to EVOH, PVDC exhibits a better barrier to oxygen and water vapor in in humid and freezing conditions. However, as illustrated by the Figure 3, the nanocellulose displays the highest oxygen barrier properties than both resins barrier, i.e., PVDC and EVOH used today [70].

Even though, EVOH is a moisture-sensitive material, manufacturers have been focused their attention on its production because it creates less environmental concerns with respect to PVDC and aluminum foils. To this regard, EVOH has a more closely similar behavior to nanocellulose (Figure 4); their gas barrier properties are compromised when exposed to high relative humidity due to the plasticization mechanism, which leads to an increase in the distance between neighboring chains (free volume) through the swelling of hydroxyl groups present in both structures [70,71,72]. Figure 4 also shows that the oxygen barrier of the CNCs is maintained up to almost 40% RH (KPO2 ≈ 0), however, after that, there is a sharp increase in O_2_ permeation, whereas the EVOH that behaves even similarly has a slightly higher but constant oxygen permeability from 0% to 50% RH, which then increases considerably as the RH increases.

EVOH is a thermoplastic material that can be coextruded, coated and laminated in multilayered structures while a neat NC is not a thermoplastic one; therefore, it must be used in combination with thermoplastics to make their application possible in packaging. The lamination is a technique that has allowed alleviating the water sensitivity of the EVOH and made possible their use in flexible packaging materials; hence, the same strategy can also be adopted for the NC applications. That being said, it becomes clear that the NC can share of a part of the market of the resin barriers such as EVOH and PVDC used in various packaging applications including for example polyethylene (PE), oriented polyethylene terephthalate (OPET), polyamide (PA), oriented polypropylene (OPP) (Table 1). NC can also replace the EVOH that has recently been explored and proven to be effective as barrier layer to oil-saturated hydrocarbon (MOSH) and mineral oil aromatic hydrocarbons (MOAHs) [75].

Prior to initiating any packaging manufacturing based on the NC, the productive capacity of the nanocellulose has to be guaranteed. According to Research and Markets (2019), the PVDC whose world production was 242 kt in 2014, compared to 142 kt of EVOH, has remained the leading technology in the world global resin barrier market even in 2019. The global EVOH market is expected to reach 1123.3 million USD by the end of 2026 (Research and Markets, 2019). By comparison, according to TAPPI (2018), the world global market of the nanocellulose is estimated to be 13,000 tons and projected to reach about 150,000 tons by 2045 (Nanocellulose Market Forecasts, 2017). From the 2015 TAPPI report, the nanocellulose market was around $250 million in 2014, with projected growth of 19% to 2019 (Table 2, Table 3, and Table 4).

## 5. Why Focus on Nanocellulose Coatings?

In this review, attention has been given to nanocellulose coatings for various valid reasons. First, it can be seen that despite extensive research through many published works, written books, and conferences about the nanocellulose, its implementation in food packaging has not taken off. According to Directive (EU) 2018/852 of the European Parliament and of the Council of 30 May 2018 amending Directive 94/62/EC on packaging and packaging waste, packaging manufacturers and operators should implement a waste management to be transformed into sustainable material management and a political and societal incentive should promote recovery and recycling as a sustainable way to handle natural resources within circular economy. Among different applications of nanocellulose described above, the coating technique appears to be the most promising and sustainable approach to accelerate the incorporation of nanocellulose into packaging industries. The casting technique could be an interesting approach to improve both mechanical and gas barrier properties of the nanocellulose; however, it is a complex operation to manage on the industrial scale of packaging manufacturing. In addition, the large amount of nanocellulose required for casting applications could make the choice less economically viable for manufacturers [69]. The coating is a well-known and developed technique which has long been used to cover functional surface in packaging for coating papers and polymers. Therefore, the emphasis placed on the coating can be justified by the fact that its application that can be rapidly extended to the industrial level taking account the low investment costs for its implementation. Furthermore, due to the lack of heat-sealability capacity, the nanocellulose must be used in association with thermoplastic polymers to manufacture a new packaging material. The most important advantages of the coating are the uniformity, preservation, and continuity of the crystalline lattice of the CNCs layer. Therefore, as indicated in Figure 5, works on NC applications revealed that the coating could provide the substrate with 5 types of barrier requirements such water molecules (liquid and gas), oxygen, carbon dioxide, volatile compounds (MOAH and MOSH), and grease properties useful for food packaging [77,78].

Other researchers have shown that the nanocellulose coatings may also be used in combination with other matrices to create active packaging. Missio and coworkers have produced nanocellulose-tannin films that have improved gas barrier properties in humid conditions and antioxidant properties upon water soaking [79]. Antimicrobial activity of about 98.8% reduction in CFU of *Saccharomyces cerevisiae*, Gram-negative bacteria *Escherichia coli,* and Gram-positive bacteria *Staphylococcus aureus* has been made possible by nanocomposites including CNCs, titanium, and wheat gluten [80]. Lavoine and collaborators have implemented new release systems for active packaging based on the microfibrillated cellulose-coated paper [81]. In a more recent work, coatings based on MFC and nisin have been incorporated with the biaxially oriented polypropylene/low density polyethylene (BOPP/LDPE) laminates to create an antimicrobial packaging. As a result, the coated laminates showed better oxygen barrier compared to the uncoated ones (24.02 vs. 67.03) and exhibited antimicrobial properties, with a growth inhibition of *L. monocytogenes* by 94% [82]. From this global research effort, a new door is opened, that of using a safe and more sustainable material like the NC in replacement with the resins gas barrier currently used in food packaging.

## 6. Optimization of NC Coatings for Packaging Applications

When it comes to nanocellulose coatings and its adhesion with the substrate, some important factors such as (1) type of the nanocellulose, (2) neat or modified nanocellulose, (3) type of substrate, (4) type of surface treatment, (5) type of coating process, and (6) coverage and thickness of the coating can be considered (Figure 6). After the coating with the nanocellulose, the gas barrier of the coated material improves significantly. However, the preparation and coating phases of the nanocellulose are very critical operations, since any lack of precision leads to a sharp increase in the permeation of the gases through the coated material. This chapter focuses exhaustively on the most important precautionary measures to consider for the optimization of the NC coatings.

### 6.1. NC Redispersion

To reach their end-use destination, NC suspensions should be dried to make their shipping and transportation practical and sustainable. The form of dried NC represents a critical factor for coating applications because their complete redispersion requires a particular attention. That being said, NC can be commercialized in various forms: never-dried suspensions, freeze-dried, oven-dried, spray-dried, and supercritically dried. Importantly, the dried NC should be kept at low temperature and humidity while the wet NC should be kept in the refrigerator at 4 °C. Drying NC has an important relevance on the organization, morphology, conformation, and crystallinity [83]. Spray-drying technique is more employed on an industrial level because it is more practical and sustainable in terms of time and costs of the operation. On the other hand, freeze-drying creates flake-like and iridescent CNCs bundles and hence, the complete breakdown of their aggregates is a complex and energy-consuming operation including sonication and handheld homogenization to obtain a complete NC redispersion. In the same way, a protocol proposed by Beck et al. for the redispersion of spray-dried CNCs must be scrupulously followed [84]. For brevity, it is highly recommended to put the CNCs in water (small amount added gradually) and possibly in presence of sodium chloride or polyethylene glycol in conditions of vigorous stirring for at least 1 h for favoring a complete redispersion of all the CNCs aggregates. After, it can be followed by 10–20 kJ sonication energy per gram of CNCs [85]. The temperature of the dispersion must be controlled during the sonication in an ice bath not exceeding about 50–60 °C [55].

Due to the greater entanglement, MFC/CNFs are not commonly dried because their redispersion is complex and energetic. However, a successful redispersion of dried carboxymethylated CNFs and mixture of CNFs and carboxymethyl cellulose in water has been reported [86]. Missoum et al. used an inorganic salt like the sodium chlorite to block hydrogen bonding formed during the MFC/CNFs freeze-drying and thus avoiding the hornification of the structure and facilitating the redispersion in water [87].

### 6.2. NC Surface Chemistry

CNCs surface charges have to be considered because not only they dictate the redispersibility of the NC but they also affect the adhesion with the substrate to be coated (i.e., papers and plastics). CNCs obtained by HCl hydrolysis do not contain charges on their surface; therefore, the pH adjustment is not necessary. Moreover, CNCs-HCl are not suitable for coating because they are less redispersible even with intense sonication due to van der Waals interactions and hydrogen bonding present [18]. The other types of CNCs and CNFs suspensions are pH dependent due to protonation and deprotonation of their charges and consequently, their pH must be adjusted before the coating for obtaining better adhesion with the substrate. More specifically, the pH of carboxylated NC suspension should be 6–8, whereas that of CNCs with sulfate half ester groups should be >4.3 and neat MFC/CNFs could have a pH < 12.3 [74,88].

### 6.3. NC Concentration

The rheological properties of the NC suspensions dictate the concentration to choose for coating. The viscosity of the suspension strongly depends on type of extraction and type of the cellulosic raw material. To illustrate that, at the same concentration and extraction type, CNCs obtained from wood pulp exhibit higher viscosity with respect to CNCs obtained from cotton linters and oxidized CNCs (APS, TEMPO, etc.) show lower viscosity compared with uncharged CNCs (HCl). For coating deposition, the concentration between 6 and 8 wt% of CNCs slurry has been adopted in many works [38,78]. As a rule of thumb, a lower concentration requires several CNCs layers until it reaches the necessary thickness for a complete coverage of the material, whereas a much higher concentration causes the brittleness of the CNCs coating after drying. However, to mitigate the CNCs brittleness, plasticizers such as the sorbitol and glycerol have been employed. On the other hand, MFC/CNFs have a higher tendency for gel formation at low consistency; therefore, the highest concentration for MFC/CNFs coating should be 3 wt% and more than a layer would be needed for a complete coverage of the substrate. Due to differences in features like size, morphology, aspect ratio, surface chemistry, and affinity with the substrates, the growth of CNCs and MFC/CNFs layer is not similar during the deposition of multiple depositions. For CNCs deposition, the thickness increment per bilayer was found between 2 and 39 nm, whereas for CNFs, below 20 bilayers deposited, the thickness growth was found between 2 and 12 nm depending on the type of coating and the substrate to be coated [89,90].

### 6.4. Coating Process and Storage of Coated Materials

Due to the hydrophilic character of the NC, substrates to be coated must be subjected to corona, plasma, and UV treatment for increasing their surface energy and adhesion with the aqueous NC suspension. The intensity of the treatment (power and time) to be set depends on the initial hydrophobicity of the substrate, which can be known from the evaluation of the water contact angle noting that a high hydrophobic material requires an intense treatment. Following coating, the coated plastic films are suspended and dried under lower humidity and temperature conditions (30 °C/50% RH) for about 48 h. The coating technique has been demonstrated to play an important role in the obtainment of a certain thickness and weight of the nanocellulose layer on the substrate. Coating techniques such as bar/rod coating, size press, and spray have been widely used to coat the nanocellulose suspension even in multilayer structures. Actually, a bar coating deposit weigh of 1 g/m^2^ of the CNCs in a single layer application and a thickness of 1 µm were employed [38]. It was also reported that for bar coating, the number of layers is proportionally linked to the thickness. As a matter of fact, 5- and 10-layer coating of slurry of MFC at 2% provide weights of 6–7 and 14 g/m^2^, respectively. On the contrary, the press coating was shown to be not really effective to obtain higher thickness [63,91]. In fact, after 5-layer coating of 3.5–10% of MFC on bleach paper, the weight was only 2–8 g/m^2^ because of the pressure applied by the rolls during the coating [81]. It is also important to note that the drying of the coating has been reported to influence the layer thickness. For instance, it was shown that contact drying is very often used for coated plastic films, whereas IR drying is more suitable for coated papers [92]. Koppolu et al. successfully used the roll-to-roll coating technique with noncontact infrared drying and air drying to coat CNF onto paper at speeds of 30 m/min [93]. In another work, it has been shown that by increasing the rod metering speed and the drying temperature, the thickness and the weight of the coating decrease; in addition, it has also been reported that the WVTR of polymer dispersion coatings decreases with the increase of the drying temperature of the coating [94,95].

## 7. Practical Solutions for the Protection of NC Coatings

Nowadays, an ideal food packaging must meet all the requirements of food safety and comply with environmental concerns concomitantly. One of the strategies to implement a food package that encompasses all consumer needs is to resort to eco-friendly laminates that combine multiple layers of materials with different functions in terms of gas/oil/water/aroma barrier and mechanical properties.

Nanocellulose-based coatings have been demonstrated to be very effective in the improvement of gas barrier properties of papers and plastic films. Improvements of gas barrier properties of plastics and papers containing cellulose nanomaterials as components have been widely proven in many works [44,96]. On the other hand, Fotie et al. have shown that the excellent gas barrier is obtained under dried conditions but can be completely lost in higher relative humidity (RH) and therefore, inappropriate for food packaging [74]. Several attempts of chemical modifications to provide hydrophobic nature to the nanocellulose have been partly discussed in this review. Yan et al. successfully produced hydrophobic microfibrillated cellulose using alkyl ketene dimer (AKD). MFC was solvent exchanged by ethanol and ethyl acetate and mixed with AKD in the same solvent at 130 °C for 20 h, and the water contact measured on hydrophobic MFC pellets was higher than 100° [83]. Even though, the hydrophobization of the nanocellulose seems to increase its hydro repellent properties, it can be noted that it does not always improve gas barrier properties. In this respect, one of the best ways of implementing nanocellulose coatings, which exhibit both water vapor and gases (O_2_ and CO_2_) barrier, required in food packaging should be a double-layer coating having as first layer (confined) pure nanocellulose for gas barrier and as second layer (exposed), hydrophobic nanocellulose or other water-repellent material (very thin aluminum layer) for water vapor barrier (Figure 7). Österberg and coworkers obtained a water-resistant multilayered structure based on CNFs and wax coatings having the wax layer as an outer layer to insulate the CNFs layer against the humid ambient [96]. Acetylated microfibrillated cellulose with high hydrophobic behavior has been produced without modifying the morphology and the tensile strength of the structure [97]. Yang also achieved superhydrophobic cellulose nanocrystals having a water contact angle of 150° by acrylamide grafting [98]. A water-resistant cast-coated paper has been created with CNFs/PLA composites, which has lower WVTR (34 g/(m^2^day) compared to the control value (1315 g/(m^2^day) at 38 °C/90% RH [99]. In 2011, Korhonen prepared by freeze-drying highly porous nanocellulose aerogels from microfibrillated cellulose and by functionalizing with titanium dioxide; the matrix obtained was oil absorbing and capable of floating on water [100]. Kisonen set up a very hydrophobic CNFs films after they were coated with alkylsuccinic esters of the hemicellulose [101]. A hydrophobic CNFs has been implemented through a very easy approach. In that work, CNFs films were coated with a silicone filament by polycondensation to introduce reactive vinyl groups for a UV-induced functionalization with perfluoroalkyl thiols [102]. In view of the first approach, the biocomposite (coated material) can be created in combination with a sealable biopolymer like PLA film creating a compostable packaging or with sealable synthetic polymers like PE and PP creating bio-based packaging as well. This approach is worth experimenting because it appears feasible and realistic since the package can be fabricated and sealed with ease (Figure 7).

## 8. Lamination of Coated Materials

Technically, the implementation of the nanocellulose coating can be swift with the fabrication of multilayer structures in which the coatings are confined between laminated hydro repellent polymers like PE, PET, and PP (Figure 8). The hydrophobization of the nanocellulose might be irrelevant in this case if the water barrier is actually provided by synthetic polymers. The advantage of such strategies is that it is possible to coat the polymer with neat nanocellulose and subsequently, laminate it to obtain a bio-based packaging like PET/NC/PE, or PET/NC/PP and PE/NC/PE can be manufactured on the industrial scale where coating and lamination have long been used. Furthermore, it might be possible to create fully compostable laminates structured, e.g., of cellophane/aluminum/nanocellulose/PLA respecting the thickness requirement of the aluminum [103].

Such strategies have been successfully implemented by setting up bio-based and fully compostable laminates based on the cellulose nanocrystals. Very recently, it has been shown that through lamination, it was possible to create multilayer materials based on NC layers. Fully compostable laminates structured of CNCs layer, cellophane and metalized aluminum (<1 µm), and PLA showed fourfold superior gas barrier properties (O_2_ and CO_2_) with respect to synthetic laminates composed of EVOH layer, PE, and PET at 40 °C and 35% RH [104]. Wang et al. used such approach to mitigate the water sensitivity of casted CNCs and CNFs films whose thickness was 36 and 39 µm, respectively. After the lamination of the casted nanocellulose films with the polypropylene film and polyurethane adhesive, the water vapor transmission rate dropped from 516 to 1 g/(m^2^day) as well as the oxygen transmission rate dropped from 126 to 6.1 cm^3^/(m^2^.day.bar) at 80% RH [69]. In another study, similar strategy has been adopted for an immediate and swift application of the nanocellulose in packaging even in higher RH (50–80%) by resorting to lamination. As a result, the oxygen permeability performed on the laminates based on CNCs coatings was null at 80% RH [38].

## 9. Nanocellulose in Food Packaging Applications: MFC/CNFs or CNCs?

Food packaging manufacturers have been reluctant in the past years to incorporate the nanocellulose in their products. Therefore, they need to build up a certain guarantee based on reliable information and to be reassured about the successful implementation of the nanocellulose in food packaging.

### 9.1. Yield and Performance

In terms of physical and chemical features, CNCs and MFC/CNFs are different as shown above (introduction). However, they can exhibit similar performance; in particular, as a barrier to gases, grease, or aroma or if used in combination with other compounds, they can allow to create an active packaging. Based on the raw materials and the type of process adopted for the extraction, CNCs yield varies from 50% to 65%, whereas that of MFC is about 90% [14,105,106,107,108,109]. As pointed out, the coating approach seems much closer to the manufacturing of the nanocellulose-based packaging on an industrial scale. The coating of the concentration of CNCs slurry can even reach 10%, whereas that of the MFC/CNFs cannot exceed 3% because they have a different viscosity and rheological properties. To demonstrate similar performance, the thickness of coating layer of the CNCs can be less than 1 µm while that of the MFC must be at least 2–3 µm [78,110,111]. Although still under discussion, MFC has been shown to exhibit a better oxygen barrier than CNCs. However, although the MFC yield is much higher, more quantity in needed to offer the same barrier performance. In contrast, by comparing castings of the same grammage (g/m^2^) based on CNCs and CNFs/MFC extracted from softwood kraft pulp, Wang et al. revealed that laminates including MFC casting exhibited better oxygen barrier properties while those including CNCs casting showed higher transparency. In addition, the castings thickness of the laminates was between 14 and 39 µm corresponding to 20.5 and 50.1 g/m^2^, respectively. It is important to note that a much higher NC thickness displays a higher swelling capacity if exposed under humid conditions and as a result, the gas barrier properties will be compromised. In terms of comparison of optical properties either in coating or castings, materials (plastics and papers) including CNCs are more transparent than those containing MFC/CNFs. In several works, the mechanical properties of both materials have been studied. MFC/CNFs castings showed higher mechanical properties with respect to CNCs ones whose structure is more brittle [69]. Furthermore, multilayer laminates including thicker nanocellulose castings exhibit higher tensile properties than the thinner ones. The WVTR is high in coating and casting of the nanocellulose, hence, the protection of the NC layer through the lamination is needed. A notable advantage was given by a very low oxygen permeability of CNF-coated paper at 0% and 40% RH due to inter- and intrahydrogen bonds between the -OH group fibrils and chains of individual microfibrils and high adhesion between the paper and the CNFs chains. The layer thickness was 5 g/m^2^; however, that oxygen barrier dropped dramatically at 65–80% RH due to the CNFs sensitivity to humidity. The air permeability of the coated paper decreased significantly with an MFC coating in agreement with the oil barrier properties it showed [112].

### 9.2. Regulations about CNCs and MFC/CNFs

Cellulose-based materials have generally been safe for food packaging applications. However, when it comes to nanomaterials (NMs), particular regulations have to be respected. According to EU Recommendation 2011/696, NMs are intended as a natural, incidental, or manufactured material containing particles, in an unbound state or as an aggregate or as an agglomerate and where, for 50% or more of the particles in the number size distribution, one or more external dimensions is in the size range of 1–100 nm [113]. Even though the cellulose nanomaterials (CNMs) are of significant scientific research and economic interest, it remains that the European commission has not yet authorized them as food contact materials (FCMs). In 2011, the European commission established that the manufacturing of FCMs “substances in nanoform should be used only if explicitly authorized” [114]. As such, it has to be acknowledged that there is a set of inorganic nanomaterials (i.e., titanium nitride, zinc oxide, and silicone dioxide) that has already been approved by EFSA and currently used as FCMs [115].

The reason is that there are doubts and voids of knowledge about the nanoparticles impact on the human health. Recent investigations on the harmonization of the analytical approaches and risk assessment of the nanoparticles have been carried out. A recent work demonstrated that cellulose nanoparticles have always been part of our daily intake. The findings provided further necessary and preliminary information/data to the EFSA for the approval of the cellulose nanoparticles as FCMs [116]. Although studies have shown that, for 10 nm diameter particles, an apparent diffusion coefficient (D) of 1.1 × 10^−35^ cm^2^ s^−1^ theoretically calculated in a LDPE host matrix exhibits a benign risk of inadvertent migration [117], it remains that the same investigation has to be performed specifically on CNCs to confirm those results. Consequently, when it comes to decide which of CNCs and MFC/CNFs should be incorporated into food packaging, it becomes clear that according to the existing EU regulation, for immediate use as food contact materials, only MFC can be used for European food packaging.

### 9.3. Processability Performance and Production Costs

It is important to consider the aspect of processability, energy consumption, and efficiency when it comes to packaging manufacturing. The paradigm shift from synthetic polymers containing EVOH or PVDC to polymers containing the nanocellulose requires further investments. An immediately complete replacement of barrier resins (EVOH and PVDC) is not possible for various reasons including higher production cost of the nanocellulose and insufficient production to cover all the demand of EVOH and PVDC packaging market. However, the cost of manufacturing of packaging-based NC can be alleviated as the same coaters and laminators actually used for packaging manufacturing can be used. It probably needs more machines but the processing line may remain the same as that of some types of applications with barrier resins (EVOH and PVDC). It is evident that casting requires a much bigger consumption of the nanocellulose compared to the nanocellulose coating whose thickness can be between 1 and 3 µm [62,65]. More importantly, when comparing CNCs and MFC rheological properties, the MFC suspension exhibits much higher viscosity compared to CNCs suspension. Actually, about 1 µm layer of CNCs coated at 6 wt% displayed the required performance [65]. To obtain similar performance, more than an MFC layer will be required to obtain the ideal thickness because it is very difficult to handle more than 3 wt% of MFC slurry (too viscous) for coating. As a result, in terms of energy consumption and production, CNCs coating will be more sustainable than that of MFC. There is a little information about the production cost and the price of the nanocellulose. However, few researchers have evaluated the costs of the CNF production in relation with the type of the pretreatments. For the same raw materials, the most expensive CNFs were TEMPO-CNF (205.73 €/kg) and the cheapest CNFs were those obtained from mechanical pretreatment (2.25 €/kg) followed by acid-CNF (7.33 €/kg) and enzymatic-CNF (13.66 €/kg) [118]. The cost of conventional CNCs production stage (sulfuric acid) was found to be 1.54 $/kg, whereas by using a water subcritical treatment, the cost could have dropped to 0.02 $/kg [119]. Strategies of containing the cost of production of the CNCs from wood pulp have been investigated. A recent work showed that, to reduce the cost of production of the CNCs, it is necessary to optimize and make efficient the step of dissolving pulp since the price of the lime and the sulfuric acid is almost constant over the years [120]. Even though the price of the nanocellulose is usually around 4–10 $/kg, trends of the price in relation with the type of the NC can be observed from a recent study [121]. On one hand, the price of the MFC/CNFs can be predicted to be lower than that of CNCs. On the other hand, even though steps such as fractioning, refining, and high-pressure homogenizing to produce the MFC/CNFs require high energy to break down fibers into fibrils, the production and yield are much higher with respect to CNCs. Both CNCs and MFC/CNFs can be commercialized in various forms including freeze-dried, never-dried, oven-dried, spray-dried, and supercritically dried.

### 9.4. Biodegradability and Safety

Several scientific works have proven the biodegradability of the nanocellulose. Coma et al. conducted the first study of the biodegradability of the nanocellulose used as a reinforcer in the polymers, and according to their findings, the complex matrix evaluated was biodegradable. Although the nanocellulose is a very tiny and small particle, its biodegradability is not compromised. In fact, they reported that the nanoparticles from cellulose and starch were even rapidly degraded than their macroscopic counterparts. It is strong evidence that the nanocellulose incorporated into the packaging can be easily recyclable [122,123]. Nanomaterials have recently drawn the attention of the scientific community to their possible adverse effects on health, and for that, their safety has extensively been evaluated accordingly. Some researchers of the field have evaluated potential cytotoxicity, genotoxicity, and ecotoxicity of the nanomaterials. Nanomaterials properties differ from their parent bulk materials because of their smaller size, different morphology, and larger surface area resulting in their ability to cross natural barriers including electronic and plasmonic effects. Pure cellulose cellulose-based foods additives (i.e., carboxymethyl cellulose and microcrystalline cellulose) are generally known as a safe and nontoxic food substance. CNCs have been assessed by the potential environmental risks of the carboxymethyl cellulose and cellulose nanoparticles, and according to their findings, the toxicity potential and environmental risk of both cellulose-based materials are negligible [124]. Moreira and coworkers tested the potential genotoxicity of bacterial cellulose on fibrous nanoparticles through in vitro analysis and other techniques and no genotoxic effects were found [125]. Vartiainen found no inflammatory effects or cytotoxic on mouse and human macrophages after being exposed for 6 and 24 h to nanocellulose [126]. DeLoid et al. employed in vitro approach to assess the effects of a 24 h incubation with tract simulator-digested CNFs and CNCs at 0.75% and 1.5% w/w on the cytotoxicity, cell layer integrity and oxidative stress through a triculture of small intestinal epithelium. According to their findings, no significant changes in cytotoxicity, reactive oxygen and monolayer integrity was observed. In addition, in vivo toxicity performed on rats gavaged with 1% w/w CNFs suspensions showed no significant differences in hematology, serum markers, or histology between controls and rats given CNFs [127]. Functionalized nanocellulose (carboxylated) has been reported as nontoxic substances. It was assessed by using standard ecotoxicological and mammalian test protocols and have, to date, been shown to be practically nontoxic in each of the individual tests [128]. In addition, CNCs have recently obtained regulatory approval under Canada’s New Substances Notification Regulations (NSNR, 2012) for unrestricted commercialization and use in Canada. Recently, Environment Canada has also partnered with the National Research Council and academic laboratories of Canada to initiate a project on the development and support of test methods to characterize nanomaterials for the purposes of regulatory identification and to support risk assessments.

## 10. Summary and Conclusions

Table 5 reports various ways of utilization of the cellulose nanocrystals (CNCs) and cellulose micro/nano fibrils (MFC/CNFs) for packaging requirements. As indicated, below 40% of relative humidity (RH), the coating and the casting of the nanocellulose can offer an outstanding barrier to gases, MOSH, and MOAHs. However, relative humidity superior to 40% compromises the initial properties of the nanocellulose; hence, it is necessary to resort to effective solutions including hydrophobic NC and/or lamination to protect its layer against the humid surroundings.

Table 6 highlights some of the recent works about the techniques adopted for the nanocellulose applications in food packaging. The casting/coating of the nanocellulose used in combination with the lamination seems very effective in mitigating the effect of the humidity even at 80% RH.

In conclusion, food packaging operators could successfully manufacture food packaging materials based on nanocellulose coatings on an industrial scale taking into consideration these suggested relevant guidelines:(1)MFC and CNFs can be regarded as an effective and competitive alternative to packaging including resins barrier like EVOH and PVDC.(2)Nanocellulose coatings of plastic film/papers and castings can be used for barrier to MOSH and MOAHs, gases, and oil; however, the coating technique seems more practical and sustainable for food packaging manufacturing on an industrial level.(3)The hydrophobic nanocellulose seems not to be effective in blocking the water vapor in standard conditions (90% and 38 °C).(4)Until now, the nanocellulose has not yet been made heat sealable; therefore, it must be used in combination with thermoplastic films or sealant layers to create packaging materials.(5)If the packaging is made for the storage of oil or dry products (aw < 0.4) with a high content of fatty acids food in dry environment, the coating of the nanocellulose may not require any protection.(6)If the one between the product to be conserved and the storage environment is humid (>40% RH), hydrophobic nanocellulose coatings or neat nanocellulose confined in multilayer structures are needed to preserve the integrity of the NC in humid conditions.(7)It is possible to implement fully compostable and bio-based multilayer’s packaging incorporating the neat nanocellulose; in addition, the bio-based laminates may include hydrorepellent films such as the PP, PE, or OPP to protect the coatings from the humidity.(8)Castings and coatings of CNCs are clearer and more transparent than those based on MFC.(9)On the one hand, certain modified nanocelluloses may not be suitable for food packaging applications if the chemical modification makes the NC less biodegradable/nonbiodegradable or if the modifications are implemented with unhealthy, toxic, or dangerous chemical compounds.(10)Although the CNCs coating appears more practicable than MFC one, only the latter is currently approved by EFSA and therefore can be used in food contact materials for the European Union packaging market.

## Figures and Tables

**Figure 1 nanomaterials-10-01726-f001:**
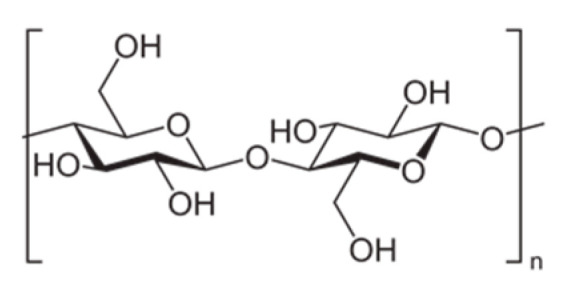
Cellobiose unit.

**Figure 2 nanomaterials-10-01726-f002:**
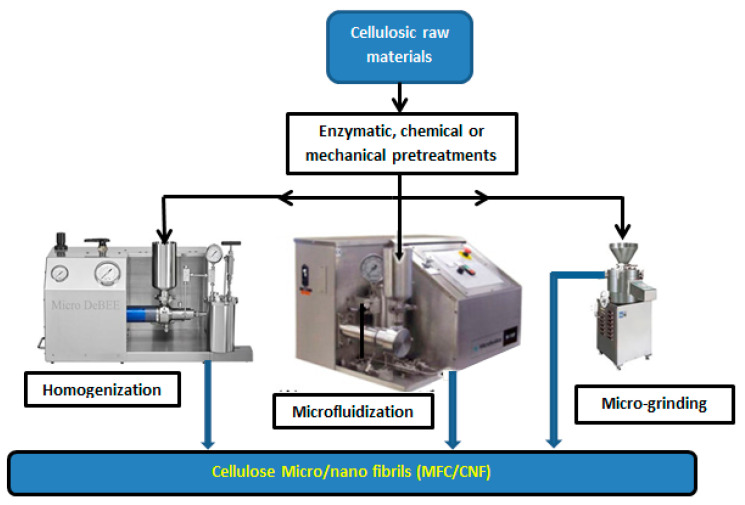
Phases for microfibrillated cellulose (MFC) production.

**Figure 3 nanomaterials-10-01726-f003:**
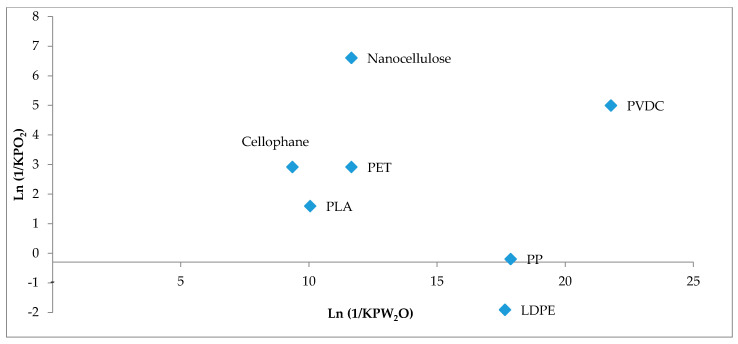
Oxygen and water vapor resistance coefficients of the nanocellulose in comparison with synthetic, bioplastics, and bio-based structures with KPO_2_ and KPW_2_O expressed in cm^3^ µmPa^−1^ day^−1^ m^−2^ and cm^3^ day^−1^ m^−1^ Pa^−1^, respectively (modified) [70].

**Figure 4 nanomaterials-10-01726-f004:**
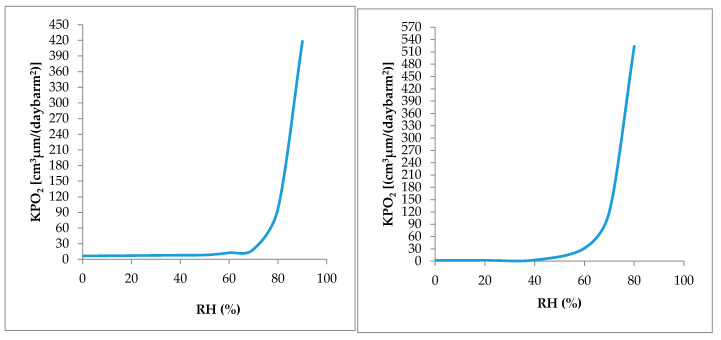
Effect of relative humidity on oxygen permeability coefficients of ethyl vinyl alcohol (EVOH) films (**left**) and cellulose nanocrystals (CNCs) films (**right**), (modified) [73,74].

**Figure 5 nanomaterials-10-01726-f005:**
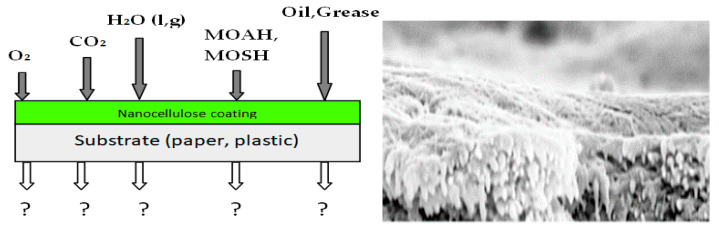
Five types of barrier provided by the nanocellulose (NC) coatings and cross-section SEM of coated polymer.

**Figure 6 nanomaterials-10-01726-f006:**
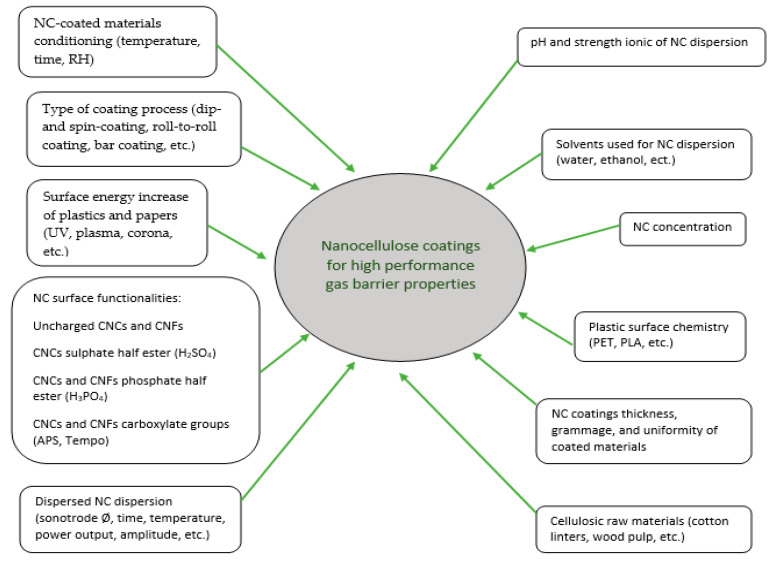
Critical factors for optimization of nanocellulose coating.

**Figure 7 nanomaterials-10-01726-f007:**
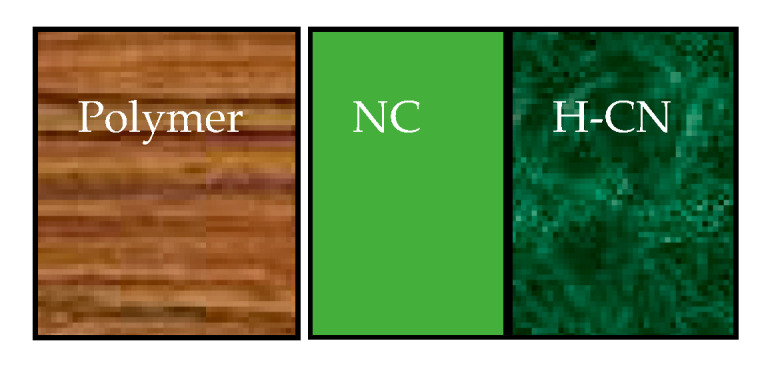
Polymer coated with neat NC and hydrophobic NC (H-NC).

**Figure 8 nanomaterials-10-01726-f008:**
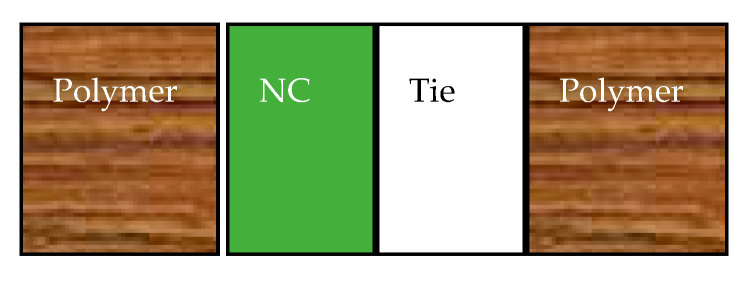
Multilayer structures: coating and lamination of nanocellulose.

**Table 1 nanomaterials-10-01726-t001:** Packaging based on ethyl vinyl alcohol (EVOH) for food applications (adapted) [76].

Structures	Requirements	Applications
PE/paper/tie/EVOH/sealant	Gas and flavor barrier, pinhole resistance	Juices, jam, snacks
PE/paper/tie/EVOH/tie/EVOH	Gas and flavor barrier, pinhole resistance	Juices, soups
PA/EVOH/PE	Gas and flavor barrier	Jam, raw meat
OPET/tie/EVOH/tie/sealant	Gas barrier, transparency	Snacks, lid stock
OPP/tie/EVOH/sealant	Gas and flavor barrier, water barrier	Snacks, spices, jam, rice paste

**Table 2 nanomaterials-10-01726-t002:** World production of cellulose nanofibrils, 2018 (per producer).

Producers	Process	Capacity (tons/year)
Asia	Modified hydrophobic, oblique collision, TEMPO, phosphate esterification, aqueous counter collision	754
Europe	Chemical, enzymatic	10
UK	Chemical pretreatment	100
USA	TEMPO, SO_2_ fractionation, chemical	391

**Table 3 nanomaterials-10-01726-t003:** World production of microfibrillated cellulose, 2018 (per producer).

Producers	Process	Capacity (tons/year)
Asia	High-pressure homogenizer, TEMPO	200
Europe	Enzymatic, chemical, and mechanical	1635
UK	Mechanical and minerals	8800
America	Mechanical	25

**Table 4 nanomaterials-10-01726-t004:** World production of cellulose nanocrystals, 2018 (per producer).

Producers	Process	Capacity (tons/year)
Asia	Unmodified and modified, proprietary	Pilot
Europe	Enzymatic, chemical hydrolysis	35
America	SO_2_ fractionation, reactive extrusion	130
Canada	Sulfuric acid hydrolysis, catalytic conversion	267

**Table 5 nanomaterials-10-01726-t005:** Nanocellulose applications for packaging requirements.

Packaging Requirements		ONLY Coating (i.e., LbL)	ONLY Casting	Coating and Laminating	Casting and Laminating	Extrusion-ES
O_2_ and CO_2_ barrier	0% RH	+++	+++	+++	+++	+
	>40% RH	+	+	+++	+++	-
Water vapor barrier		−	+	+++	+++	+
Grease barrier		+++	+++	+++	+++	++
MOSH and MOAHs barrier		+++	+++	+++	+++	+
Antimicrobial activity		+++	+/-	+/-	+/-	+/-
Mechanical and thermal properties		+/-	+++	+/-	+++	+++
Transparency	CNCs	+++	+	+++	+/-	-
	MFC/CNFs	+	+/-	+/-	-	-
Regulation requirements	CNCs	-	-	-	-	-
	MFC/CNFs	++	++	++	++	++
Low production cost and less time consuming	CNCs	+++	+	+++	+/-	-
	MFC/CNFs	++	+	++	+/-	-

**Table 6 nanomaterials-10-01726-t006:** Coated nanocellulose (NC) and laminated (*), casted NC and laminated (**); cellulose nanocrystals-ammonium persulfate (CNCs-APS) (^A^); CNCs-H_2_SO_4_ (^B^); 90% RH and 38 °C (^S^).

NC Type			NC Thickness	Conditions	KPO_2_ [cm^3^ µm/(kPam^2^day)]	WVTR (g/(daym^2^)	References
**CNCs**	PET-coating	PET-CNCs ^A^	1	50% RH, 23 °C	0.36		[38]
		PET-CNCs ^B^ PLA-CNCs^B^	1 1	50% RH, 23 °C 50% RH, 23 °C	0.55 1.13		[38]
	Laminating (*)	PET/CNCs ^A^/Tie/PE PET/CNCs ^B^/Tie/PE	1 1	80% RH, 23 °C 80% RH, 23 °C	0.0006 0.0025		[38] [38]
		Cellophane/metalized aluminum (<1 μm)/tie/CNCs ^B/^Tie//PLA	1	35% RH, 23 °C	0.0047	6.31 ^S^	[104]
	Casting	CNCs ^B^	36		20	452 ^S^	[69]
	Laminating (**)	BOPP/tie/CNCs/tie/BOPP	36	80% RH, 23 °C	10.4	0.9 ^S^	[69]
**MFC/MFC**	PLA-coating	MFC					
		MFC-TEMPO	0.1 0.4	0% RH, 23 °C 50% RH, 23 °C	1.33 18		[129]
	Casting	MFC	37.7 39	50% RH, 23 °C 80% RH, 23 °C	<0.011 11.4	407.6 ^S^	[130] [71]
		MFC-TEMPO	23.3 3.19 3.19	0% RH, 23 °C 0% RH, 23 °C 50% RH, 23 °C	<0.007 0.0006 0.85		[130] [112] [112]
	Laminating (**)	BOPP/tie/MFC/tie/BOPP	80% RH, 23 °C		42.2	0.8 ^S^	[69]

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
