# Peer review of "Manufacturing of Food Packaging Based on Nanocellulose: Current Advances and Challenges"

_nanomaterials, 2020, doi:10.3390/nano10091726_

Round 1

Reviewer 1 Report

  1. English need to be improved, and check for grammatical errors.

The following sentence is not very clear

 Nowadays, degradation and pollution of the environment due to synthetic polymers represent one the 8 biggest worldwide challenges

  1. There are several types for example: line 267  

Layer-by-layer (LbL) assembly, and 268 written as LDL) correct it to LbL

Reviewer 2 Report

the manuscripts content is good and very timely with trends toward elimination of petroleum based packaging. the authors need to have a good English speaking editor go through the manuscript to correct the use of odd phrases. Examples are line 186"strongly obstacles", line 194 "superficial capabilities", line 217"improved twice", and line 309 "casted". Although one can in most cases figure out what is intended they certainly would not be the way an English speaking person would say them.
